# Utilizing Insects as Bioindicators: An Approximation for Conservation in Urban Lentic Ecosystems in Central Chile

**DOI:** 10.3390/insects15110831

**Published:** 2024-10-24

**Authors:** Sebastián Rodríguez, Amanda Huerta, Álvaro Palma, Francisco Vicencio, Jaime E. Araya

**Affiliations:** 1Departamento de Silvicultura y Conservación de la Naturaleza, Facultad de Ciencias Forestales y de la Conservación de la Naturaleza, Universidad de Chile, Santiago P.O. Box 9206, Chile; 2Laboratorio de Criminalística Central, Policía de Investigaciones de Chile, Santiago 9031242, Chile; 3Independent Researcher, Santiago 7790562, Chile

**Keywords:** aquatic insect assemblage, water quality bioindicators, water physicochemical variables

## Abstract

This study considered using insect families as bioindicators to establish the health status of an ecosystem’s lentic bodies. The water quality in urban lentic bodies in the Metropolitan Region, Chile, was evaluated using aquatic insect family assemblages and physicochemical variables for conserving aquatic life. The eudominant insect families were Corixidae (associated with very bad water quality), Chironomidae (very bad), and Baetidae (good). The TSS (total suspended solids), P (phosphorus), and EC (electrical conductivity) were strongly positively correlated and negatively associated with DO (dissolved oxygen) levels. The TSS level was the most significant influential factor. Based on the nitrogen and phosphorus levels in the three water bodies, all of them were eutrophic. Given the diversity and complexity of ecosystems, studies should delve deeper into wetlands to establish methods that contribute to determining water quality, using insects as bioindicators and physicochemical variables. Considering the diversity of these water bodies, it is possible to determine not only the water quality but also the conservation state of the whole environment.

## 1. Introduction

A major concern of society in the face of the current climate and ecological crisis is impacts on water resources. Worldwide, climate change is predicted to affect these resources differently between regions, depending on their geographical characteristics and orographic conditions [1]. Industrial and agricultural activities impact water availability in central Chile, which has a predominantly Mediterranean climate combined with climate change and megadroughts [2,3].

Biodiversity conservation is a global issue, and the challenge is to integrate all levels of biodiversity to ensure the long-term evolutionary potential and resilience of biological systems [4]. The Chilean Standard 1.333 Of. 76 (NCh 1.333), “Water quality requirements for different uses” [5], and the technical “Guide for the establishment of secondary environmental quality standards for continental surface and marine waters” [6] regulate water use and thus protect and preserves ecosystems which are home to various species. NCh 1.333 set water quality criteria based on scientific requirements to “protect and preserve the quality of waters for specific uses, from degradation by contamination with waste of any type or origin” [5]. The “Natural quality of continental waters” is defined as “the value of the unit or value of the concentration of an element or compound in the body and/or course of continental surface water, which corresponds to the estimate of the original situation of water without anthropic intervention plus the permanent, irreversible or unchangeable situations of anthropic origin” [6]. These documents describe water quality according to its physical and chemical variables (temperature [T], dissolved oxygen [DO], and pH, among others) which are applicable to water uses such as human and animal consumption, irrigation, recreation, aesthetics, and aquatic life conservation. This traditional definition of water quality related to its uses is far from absolute but is relative to the aim of this resource. These documents do not necessarily state the natural quality of the water [7]. Among the physicochemical variables used to determine water quality, T, DO, pH, and electrical conductivity (EC) are considered critical [8].

Biological variables have been used since the end of the 20th century, as they were considered more efficient in representing the continuous events that occur in a water body [9]. Our study focuses on evaluating water quality for the development of aquatic life and highlights the limited experience in using biological variables in Latin America. Thus, greater information collection and the standardization of evaluation and monitoring methods are necessary [10].

Bioindicators are taxonomic groups or species capable of reflecting an ecosystem’s conservation state, diversity, endemism, and disturbance degree [9]. Studies on water quality bioindicators in Chile have focused mainly on the central–southern zone [10], where different biotic indices have been evaluated. These studies have focused on rivers, that is, lotic bodies of water, understood as linear ecosystems that evacuate water falling over continental masses into the ocean. This gravitational transfer dissipates the potential energy contained in the water, resulting in important modifications in the morphology of rivers. In Chile, there is a lack of studies on lagoons, that is, lentic ecosystems, understood as large volumes of stored water with long retention and a slow flow velocity [11]. Lagoons reflect phenomena and events that occurred previously in the converging rivers, carrying elements and nutrients that accumulate and may show disturbances that are not easy to detect in rivers, such as imbalances in the concentration of certain polluting elements [12].

Studies of lentic water bodies, including the presence of aquatic organisms, highlight the importance of insect assemblages due to their abundance and ecological diversity [7]. They present five basic aspects: high species richness and diversity, easy manipulation, ecological fidelity, fragility to minimal disturbances (sensitivity), and short generational temporality. They are found in almost all habitats, with a wide range of responses to disturbances like pollution and sedentary habits that form an aquatic system’s health status [13]. Among the aquatic insects used for this purpose, the most important are Ephemeroptera, Odonata, Plecoptera, Neuroptera, Hemiptera, Coleoptera, Trichoptera, Lepidoptera, and Diptera [9]. All these insects play the role of bioindicators. They are part of biological indices, which can be converted through formulas into numerical values that classify water into various qualities. There is an extensive list of biotic indices (BIs); however, many are modifications and/or adaptations. These approaches are based on the sensitivity of key groups to pollution and on the number of component groups in a sample, and estimate whether these groups have been affected by physical or chemical changes in their habitat [7,14]. These indices include the Extended Biotic Index (EBI), the British Biological Monitoring Working Party score system (BMWP) [15], the Stream Invertebrate Grade Number Average Level (SIGNAL) [16], and the Family BI (FBI), modified by Hilsenhoff (1988) [14]. These biological indicators are currently the most widely used, with applications in Chile [10]. In our study, we applied the FBI, the BMWP, and the SIGNAL in Chilean conditions with the five water quality classes according to national standards [5,6], making comparing them easier. They have been used in lentic environments in varied countries, such as the USA, Finland, Ireland, Sweden, the United Kington, Estonia, Lithuania, China [17], Poland [18], and India [19,20,21,22], among others. Also, they were designed for use by technical personnel rather than entomology experts who could be trained in family-level identification.

Lentic environment insect communities share the species diversity and trophic complexity of these habitats, reflecting the various species’ life history variability and the habitats’ complexity [23]. In Mississippi, USA, aquatic beetles were studied in lentic water bodies, illustrating the importance of habitat influencing patterns of community assembly [24]. In Japan, insect diversity in lentic environments was evaluated, emphasizing implications for their habitat [25]. Researchers in Guatemala (Central America) studied aquatic macroinvertebrates to assess the water quality of various standing water bodies. The study found that species diversity is higher in areas without human influence and that their distribution is influenced by changes in physicochemical conditions [26].

The most important wetland in Chile’s Metropolitan Region (MR) is the Batuco Lagoon (Lampa Municipality), located in the intermediate depression of the Santiago basin. This wetland has been shown to receive contaminated water inputs that directly infiltrate underground deposits, together with the anthropic impact of agriculture and livestock farming practiced in these areas [27,28].

Another characteristic water body in the urban area is the Carén wetland (Pudahuel Municipality, MR, Santiago, Chile). It is a suburban surface water body with a relatively flat geomorphology and gentle hills. Two branches branch off along one section of its course and have a series of small estuaries that flow into the lagoon in a fan shape [8].

Another important body of water in the MR of Chile is the Chada Reservoir (Lampa Municipality), a lentic body with river input intended for agricultural supply. This body of water was created thanks to works carried out under the Plan for the Construction and Rehabilitation of Small Reservoirs [29]. It should be noted that, in this case, this study contributes to the conservation of this lentic body since it corresponds to the first record of water quality, constituting a basis for future studies.

Given the uniqueness and vulnerabilities of these bodies of water, and as the first approximation to measure these qualities, the objective of this study was to contribute to the knowledge of the dynamics of aquatic insect family assemblages as bioindicators of water quality, complementing the traditional use of physicochemical variables in urban lentic bodies for the conservation of aquatic life in the MR, Chile, to strengthen the determination of the diverse classes of water quality in these bodies.

## 2. Materials and Methods

### 2.1. Study Areas

Three lentic bodies in the MR of Chile were studied: Batuco Wetland (Batuco), Carén Lagoon (Carén), and Chada Reservoir (Chada), whose UTM (19H) coordinates are 6324095 N–329375 E, 6299211 N–328236 E, and 6247419 N–347707 E; 481, 470 and 431 masl; and 275.0, 28.7, and 13.2 ha, respectively (Figure 1).

Batuco is the most important urban wetland in the MR, standing out for its aquatic birdlife. Among its main threats, strong anthropic disturbance stems from agricultural and livestock impacts and groundwater infiltration. This wetland is inserted in a scrubland and sclerophyllous forest region (Figure 2a). According to the Strategy for the Conservation of Biodiversity, it corresponds to “Priority Site No. 6”, whose total surface area is 14,778 ha. This wetland is considered a biological conservation area. It is an official protection area for the purposes of the Environmental Impact Assessment System, as well as for the Environmental Impact Assessment System’s purposes for tourist interest, where hunting is prohibited [27]. In 2021, it was declared a Nature Sanctuary by the Ministry of the Environment, protected by the State of Chile, given that a high concentration of aquatic birds characterizes it as a resting, feeding, or nesting area for these species [29].

Carén is an urban water body that has suffered agro-industrial pollution events with organic matter discharge. It is currently part of the “Parque Carén” of the University of Chile, where the Technological Center for Food Innovation (CeTA) is located. It holds a Mediterranean forest, with plant species disappearing due to urban development [30,31]. Its vertebrate fauna is composed of mostly native species of amphibians, reptiles, birds, and mammals, and is a major tourist attraction. This ecosystem has been exposed continuously to agro-industrial pollution, increasing its vulnerability (Figure 2b). This area has a temperate, semi-arid Mediterranean climate, characterized by a dry season with high radiation and precipitation and cloudiness during the winter. On the riverbank are plantations of eucalyptus (*Eucalyptus globulus* Labill), native sclerophyllous forest formations, and semi-woody and herbaceous species. The area’s fauna is composed of species of amphibians, reptiles, birds, and mammals, mostly native, and is a great tourist attraction. It should be noted that this ecosystem, representative of the sclerophyllous forest in the Mediterranean climate zone of central Chile, has suffered sporadic agro-industrial pollution events that increase its vulnerability [8].

Chada is a lentic reservoir with a fluvial contribution intended for agricultural supply. This water body was created thanks to works carried out under the Plan for the Construction and Rehabilitation of Small Reservoirs [25] (Figure 2c). Unlike the other two water bodies, this water body is the only one of an anthropogenic nature and has a defined objective: storing water resources for agricultural purposes. It is estimated that more than 2000 inhabitants of the adjacent town benefit from this reservoir, especially farmers in the area, as it provides an irrigated surface of 500 ha of agricultural production, mainly fruit and vegetable plantations [29]. The vegetation includes native species of spiny forest (Fabaceae), such as *Vachellia caven* and *Neltuma chilensis*, and sclerophyllous forest, such as *Maytenus boaria* (Celastraceae), among other shrubs and herbaceous plants.

### 2.2. Sampling and Identification of Aquatic Insects

In each water body, four areas were chosen, representative of their diversity, each with a sampling station (SS) established randomly. These SSs were GPS (Garmin) georeferenced, using UTM coordinates. Sampling was conducted in two seasons and in a series of three and two years, spring (2015, 2017, and 2018) and fall (2016 and 2018) (Figure 1), except in Batuco, which had a series of in spring 2017 and fall 2018.

Aquatic insect sampling was carried out in the same seasons and series of years indicated for each water body, taking three samples of aquatic insect specimens for each SS (Figure 1). Sampling was conducted by immersing a 1 L (8 cm diameter, 20 cm length) sterile polypropylene jar with a sealable lid to quantitatively sample a known area in the water body, according to the Official Chilean Water Quality Standard NCh-ISO-5667/4:2016 [32]. The samples were taken at 20–100 cm of the edge of the water body, at no more than 50 cm deep. Next, the contents were filtered with a piece of tulle cloth attached to a plastic strainer, and the retained solid material was transferred to another 250 mL polypropylene container with 70% ethanol. Then, the samples were transferred to the Forest Entomology Laboratory, Faculty of Forestry Sciences and Nature Conservation, University of Chile, in Santiago, where they were again filtered with a 250 µm mesh sieve [8]. The samples were then cleaned, separated, and identified in Petri dishes under a stereoscopic magnifying glass and dissecting forceps, together with updated identification keys for the main families of aquatic insects [33,34]. The richness (average number of families/L) and abundance (average number of individuals/L of each family) levels were determined from the insects already identified. A non-parametric Kruskal–Wallis (H’) statistical test was used, followed by Dunn’s multiple comparison tests to identify differences between SSs (*p* < 0.05). A Mann–Whitney (W’) test was used to compare seasons with the InfoStat [35] statistical analysis program. Each water body’s relative abundance (%) of aquatic insect families was applied for each SS, season, and year. According to their rank–abundance distribution, all individuals of single families by water body were divided according to Engelmann’s (1978) [36] species abundance classification criteria, into eudominant (32.0–100.0%), dominant (10.0–31.9%), subdominant (3.2–9.9%), recedent (1.0–3.1%), subrecedent (0.32–0.99%), and sporadic (<0.32%) taxa.

### 2.3. Determination of Water Quality Classes by Biotic Indices

Each SS of the three water bodies, by season and year, was classified according to its quality with the three indices for comparison, adapted for Chile by Figueroa et al. (2007) [10]: ((a) The British Biological Monitoring Working Party score system (BMWP). Its application consists of identifying the families of a sample and assigning a tolerance value and addition, rated following the following classification: Class I, very good: >100; Class II, good: 61–100; Class III, regular: 36–50; Class IV, bad: 16–35; and Class V, very bad: <15). ((b) The Stream Invertebrate Grade Number Average Level (SIGNAL). This is an adaptation that considers the value of the BMWP and divides it by the total number of families in the sample, rated following the following classification: Class I, very good: >7; Class II, good: 6–7; Class III, regular: 5–6; Class IV, bad: 4–5; and Class V, very bad: <4). (c) The Family Biotic Index (FBI). Water quality value results from FBI = 1/N Σ ni* ti, where ni is the number of individuals in a family, ti is the tolerance score of each family, and N is the total number of individuals in the SS sample. Families are assigned a tolerance number from 0 to 10 pertaining to that group’s known sensitivity to organic pollutants: 0 being the most sensitive, and 10 being the most tolerant. This method estimates whether they have been affected by physical or chemical changes in their habitat [7,14]. The resulting SS with this formula was located in one of the five water quality classes (modified from MOP-DGA (2010) [37]): Class I, very good: 0.00–3.75; Class II, good: 3.76–4.63; Class III, regular: 4.64–6.12; Class IV, bad: 6.13–7.25; and Class V, very bad: 7.26–10.00.

### 2.4. Measurement and Determination of Water Quality Classes Using Physicochemical Variables

In parallel with insect sampling, physicochemical water variables were measured in situ in 1 L samples of the same SS with a portable multiparameter (WTW Multi 340i, Weilheim, Germany) that provides instant results (pH, electrical conductivity [EC, µS/cm], total suspended solids [TSS, mg/L], temperature [T, °C], and dissolved oxygen [DO, mg/L]). The historical T average per season (fall or spring) was determined from all the values by SSs in each water body. Then, the difference between the historical T average and that measured in situ by the SS was determined. In each SS, in turn, three subsamples were obtained in 50 mL Falcon-type tubes that were taken at 4 °C for a subsequent laboratory analysis of the total values of phosphorus (P, µg/L) and nitrogen (N, µg/L), according to the methods indicated in the Official Chilean Water Quality Standard NCh 411/1-4 [38]. These last analyses were conducted in the Environmental Chemistry Laboratory of the Ecology and Environment Section of the Central Criminalistics Laboratory (LACRIM Central) of the Chilean Investigative Police (PDI), Santiago, Chile. All the values obtained in situ and in the laboratory were compared with references in the Guide [6] and the Official Chilean Standard NCh 1.333. Of 78 [5] concerning the water requirements intended for aquatic life. Thus, each SS of each season and year was classified according to a water quality class: exceptional, first, second, third, and fourth.

### 2.5. Global Analysis of Physicochemical and Biological Variables

Based on the results above, Friedman and Wilcoxon signed-rank paired tests were used to determine the differences between biological indices, and a Canonical Correspondence Analysis (CCA) was performed with the seven physicochemical variables (T, pH, EC, TSS, DO, P, and N) and aquatic insect families, with all the data for the three lentic bodies. For this analysis, the PAST (Paleontological Statistics) software package for education and data analysis version 4.17 was used [39].

## 3. Results

### 3.1. Abundance and Richness of Aquatic Insect Family Assemblages

In Batuco (*n*= 598 individuals), SS4 had the greatest abundance and richness, considering both seasons. SS1 had the least richness (Figure 3a,b). Corixidae, followed by Chironomidae, were the eudominant insect families found in different seasons and SSs (Table 1; Figure 4a,h and Figure 5a. In the fall, SS1 and SS4 had the least and greatest abundance, respectively (significant, H’ = 9.97; *p* = 0.0188). In the spring, the abundance in SS1 and SS2 was significant, with differences between SS3 and SS4 (H’ = 18.41; *p* = 0.0003). When considering abundance differences within the same SS across seasons, only SS1 and SS3 were significant (W’ = 6.00; *p* = 0.0238 for both SSs) (Figure 3a). Regarding family richness, there were differences between SSs in the fall (H’ = 5.86; *p* = 0.0374). Between seasons, there were differences in SS2 (W’ = 24.00; *p* = 0.0238) (Table 1, Figure 3b).

In Carén (*n*= 209 individuals), SS4 had the greatest abundance level and SS3 the greatest richness level, considering both seasons. SS2 had the least richness (Figure 3c,d). The eudominant insect families were also Chironomidae (Figure 4h and Figure 5b), and Corixidae (Figure 4a and Figure 5b), followed by Baetidae (Figure 4e,f and Figure 5b) as dominant (Table 1). The greatest abundance occurred in SS4 in the fall (2016), while the greatest family richness occurred in the same SS in the spring (2015). In the fall seasons, significant differences in abundance were found (H’ = 7.94; *p* = 0.0415). Between the different seasons, SS4 presented differences in abundance levels (W’ = 66.5; 0.0288) (Figure 3c). There were differences in the richness levels of families between SSs during the fall seasons (H’ = 7.97; *p* = 0.013). Between seasons, there were differences in SS1 (W’ = 28.50; *p* = 0.0224) and SS2 (W’ = 21.00; *p* = 0.0004) (Table 1, Figure 3d).

In Chada (*n*= 296 individuals), SS3 had the greatest abundance and richness levels, considering both seasons. SS1 had the least richness (Figure 3e,f). Only Corixidae (Figure 4a,e and Figure 5c) was the eudominant insect family, followed by Baetidae (Figure 4e and Figure 5c) in season and SSs as dominant (Table 1). The greatest family richness occurred in SS2 in the spring (2015) and SS3 in the fall (2016). In the fall, significant differences in abundance occurred between SSs (H’ = 11.60; *p* = 0.0048). In the spring, SS4 had considerably greater abundance than the other SSs (H’ = 11.78; *p* = 0.0059). Between seasons, there were differences in abundance in SS4 (W’ = 21.00; *p* = 0.0004) (Figure 3e). There were differences in family richness between SSs during the fall seasons (H = 11.22; *p* = 0.0043). Between seasons, there were differences in SS4 (W’ = 21.00; *p* = 0.0004) (Table 1, Figure 3f).

### 3.2. Water Quality from Physicochemical and Biological Variables

In Figure 6, Figure 7, Figure 8, Figure 9 and Figure 10, for each water body, SS, season, and year, the temperature variation (T), dissolved oxygen (DO), electrical conductivity (EC), total suspended solids (TSS), pH, phosphorus (P), nitrogen (N), and biological index (FBI) (Figure 7) are presented. The water quality classes were obtained from the FBI, BMWP, and SIGNAL, and from the Chilean Standard NCh 1.333 [5] and Guide [6] through DO and T (Figure 8, Figure 9 and Figure 10).

In Batuco, the SS with the greatest variation in T was SS2 in the spring of 2015, with a Class III (Bad) water quality and a rating of IV (Bad) in the Chilean standard and the FBI; however, the BMWP and SIGNAL reduced its rating to Class V (Very bad). The DO level was greater in SS1 in the spring of 2018, with an Exceptional water class according to the Chilean standard, and a Class I (Very good) rating by the FBI, but a Class V (Very bad) rating from the BMWP and SIGNAL. The greatest pH was obtained in SS2 in the fall of 2018 and SS3 in the spring of 2018, above the requirements for aquatic life in the relevant standards; however, these had a Class III (Regular) and I (Very good) rating from the FBI, while both remaining in Class V (Very bad) for the BMWP and SIGNAL. The EC was greatest in SS3 in the spring of 2018, reaching Class 1 (Good) in the standard classification and Class I (Very good) with the FBI, but a Class V (Very Bad) rating from both the BMWP and SIGNAL. Similarly, in SS3, the maximum TSS were in the spring of 2015, with values outside the norm; however, it had a Class I (Very good) rating from the FBI, but a Class V (Very bad) rating from the BMWP and SIGNAL (Figure 6, Figure 7 and Figure 8).

In Carén, the SS with the greatest variation in T was SS3 in the spring of 2017, with a Class III (Bad) water quality and a Class II (Good) rating in the Chilean standard classification and the FBI; however, the BMWP index downgraded it to quality Class V (Very bad), and the SIGNAL considered it IV (Bad). The DO was greater in SS4 in the fall of 2018, with a standard classification as Exceptional, and an FBI rating of Class I (Very good), but the BMWP and SIGNAL rated it as Class V (Very bad). The greatest pH level occurred in SS1 in the fall of 2018, not satisfying the requirements for aquatic life of the standards; however, with the FBI, a Class IV (Bad) rating was obtained, while with the BMWP and SIGNAL, it remained in Class V (Very bad). The EC was greatest in SS4 in the fall of 2016, with a Class III (Bad) classification by the standard and a Class IV (Bad) rating with the FBI, but with both the BMWP and SIGNAL rating it in Class V. The maximum TSS also occurred in SS4 in the fall of 2016, with values in Class III (Bad), coinciding with the FBI classifying it as Class IV (Bad), while with the BMWP and SIGNAL rated it as Class V (Very bad) (Figure 6, Figure 7 and Figure 9).

In Chada, the SS with the greatest variation in T was SS3 in the spring of 2015, with a Class III (Bad) water quality and a Class I (Very good) rating from the standard classification and the FBI; however, the BMWP and SIGNAL indices gave ratings of Class V (Very bad) and IV (Bad). The DO was greater in SS1 in the spring of 2017, with a standard classification as Exceptional, and an FBI rating of Class I (Very good), but the BMWP and SIGNAL rated it in Class V (Very bad). The greatest pH, EC, and TSS occurred in SS4 in the fall of 2018, not satisfying the requirements for aquatic life (pH) and with class Exceptional (EC and TSS) with the standards; although no specimens were captured, it was impossible to classify it with the biological indices (Figure 6, Figure 7 and Figure 10).

### 3.3. Global Results of the Three Water Bodies

Using the standard indices, the water bodies were generally classified from best to worst quality in the following order: Chada, Carén, and Batuco. Based on the nitrogen and phosphorus levels in the water bodies, all of them were eutrophic. According to the FBI, Chada was also the best quality water body, classified in Class I, followed by Batuco and Carén, both in Class III. Meanwhile, with the BMWP and SIGNAL, all bodies were classified in Class V, except for Carén, which was in Class IV by the SIGNAL. The Friedman test showed a significant difference (*p* < 0.0001) between biological indices. A Wilcoxon signed-rank test identified significative differences between the FBI/BMWP and the FBI/SIGNAL biological indices pairs (*p* < 0.01) (Figure 8, Figure 9 and Figure 10; Table 2 and Table 3).

The CCA triplot separated the three sites based on the significant physicochemical variables and aquatic insect family community. The CCA result of cumulative percentage variation of insect family–physicochemical variables showed that axis 1 contributed 67.2% and axis 2 contributed 19.1% of the total variability and 86.3% of the total variation between both. The long arrows of TSS, P, and EC (strongly correlated with the first axis) and the DO level is negatively associated with them. The TSS level was the most significant influential factor. In Batuco, Chironomidae and Corixidae indicated very bad water quality, consistent with the BWWP and SIGNAL classifications. The position of Carén is associated principally with the presence of Chironomidae, Corixidae, Baetidae, Coenagrionidae, and Aeshnidae, indicating principally very bad water quality, consistent with the BWWP and SIGNAL classifications. In the position of Chada, Baetidae, Corixidae, and Hydrophilidae were identified, denoting a better water quality than other water bodies. However, only the FBI could detect it, as both the BMWP and SIGNAL did not, classifying it as very poor, and the standards were variables (Table 2, Figure 11).

## 4. Discussion

### 4.1. Analysis by Water Body

#### 4.1.1. Batuco

Batuco had the highest abundance and richness of aquatic insect families compared to the other two bodies of water studied, although those obtained in the SSs closest to the main access (SS1 and SS2) were lower, where an evident anthropogenic disturbance was observed by the proximity to urban and agriculture areas, coinciding by Reyes-Morales (2013) [26] in lentic bodies in Perú; however, a meta-analysis of the spatial dataset revealed that the aquatic insect abundance level was higher in lentic bodies, but was not influenced by the level of anthropogenic impact [40]. The estimated abundance could be because the eudominant families (Corixidae [Hemiptera] and Chironomidae [Diptera]) have a wide range of tolerance to changes in environmental variables. Corixidae was represented by *Sigara* sp., a cosmopolitan genus [41]. A study of freshwater British lakes demonstrated significant correlations between the distribution of some species of this genus and EC; some live with an EC of up to 1000 µS/cm, as is the case in Chile [42]. According to the FBI, the quality classes corresponding to “regular” are mainly due to the presence of the Chironomidae (Diptera), which score 7 [10], but correspond to “very bad” for the BMWP and SIGNAL, which are more restrictive. This family is known for living for long periods in waters with low DO concentrations [43], and the samples with a low value in this variable had a high abundance of these insects (fall [2015] in SS3; spring [2015] in SS4). This family was abundant in a lentic water body in Colombia, associated with high tolerance to extreme conditions [44]. A study of Chironomidae assemblages from the Araucarian lakes in south–central Chile suggested that the spatial distribution is closely related to changes in DO levels [45]. Overall, interpretations drawn for large and diverse groups such as Chironomidae may be overly simplistic. However, general points are useful for summarizing primary associations, and the group-level information, with due caution, could be valuable [46]. Also, in the lentic bodies of Perú, this family was dominant [26], with a high tolerance to anoxic conditions, as is the case in Chile, and may maintain relatively high faunistic richness and abundance under strong anthropogenic pressure, according to a study of lentic and lotic bodies in France [47]. Corixidae and Chironomidae were associated with extremely lentic environments by studying the distribution of taxa along a lentic and lotic gradient in Italy [46].

The high DO values may result from the wind conditions and the low depth of the water body [36]. The EC and TSS exceeded the standard. This may be due to the high concentration of salts, the shallow depth, and high evaporation levels, among other factors. In this same water body, a high EC throughout the basin and the consequent saline composition of the soils were observed [48]. It is also indicated that the waters fluctuate from fresh to brackish and swampy, mainly due to the low permeability of the soils. Comparing our FBI results by SSs, the values generally indicated quite different water levels than physicochemical indices coinciding with those obtained from Poland’s lakes [18]. Also, the FBI rating of the SS closest to the main access point (SS1 and SS2) was higher, indicating anthropogenic disturbance, as observed in a lake in India [19]. The BMWP and SIGNAL both gave ratings of Class V; however, in India’s wetlands, the BMWP rating was more restrictive than the SIGNAL rating [21], in comparison with the SIGNAL rating of another study [22]. Finally, this body had the worst water quality (bad) and was assessed as being less compatible for aquatic life by standards [5,6] compared to the other bodies studied, although a lower sampling effort was applied.

#### 4.1.2. Carén

In general, the low estimated abundance of aquatic insect families coincided with the characteristics of this water body, with many of its physicochemical variables not favoring aquatic life development (pH: does not comply; Trophic level [N and P]: eutrophic; and EC and TSS: Class III: [bad]) according to the standards, but deemed “regular”, by the FBI. However, “very bad” was the rating given by the BMWP and “bad” by the SIGNAL. More restrictions were found in these last indices again. Their abundance contrasted with the richness of families present. This may reflect a total absence of individuals, and it is assumed that in the absence of insects, there are no appropriate conditions for the development of aquatic life, as the presence of Chironomidae individuals as eudominant contributes to a “very bad” quality class. Corixidae, eudominant in freshwater bodies of the central zone [10], was abundant in Carén, which could be related to its physicochemical variables, such as EC [42], however here was it lower in levels (<1000 µS/cm) than Batuco. It is possibly another species of the same genus. Baetidae (Ephemeroptera, FBI score 4) [10] was dominant, particularly *Andesiops peruvianus* (Ulmer, 1920), a small minnow mayfly with distribution throughout the Andes Mountains, but mainly in lotic environments [49]. Corixidae was associated with lotic and lentic conditions [46].

Carén presented high pH levels in all seasons, exceeding those indicated by MOP-DGA (2020) [50] and SMA (2020) [51] for equivalent SSs. However, the information obtained in those studies had as a reference the LA-1 section at a sampling station of the Colina Estuary close (1 km) to the lagoon, which could explain fluctuations in its physicochemical variables from Carén, for example, through dilution in the waters. On the other hand, the Pudahuel Series soils of this area are characterized by having a slightly basic pH of a pumicitic origin [52]. Using SS4 as a reference, whose location was close to that used by the study [51], the DO values were similar in the same season and year around this previous study, coinciding with the greatest values herein in the fall and winter. For EC, in SS4, the difference between values was greater, although in all cases, the values in our study were less than those in the study [51]. This may be because of SS used by SMA (2020) [51] corresponded to one of the lagoon tributaries, a lotic water body. Comparing our FBI results by SSs, the values generally indicated quite different water levels than the physicochemical indices here, coinciding with those obtained from Poland’s lakes [18]. The FBI value in the SS closest to the main access point (SS1 and SS2) was lower, indicating higher water quality despite the anthropogenic disturbances observed. This contrasts with what was obtained in a lake in India, where there were anthropogenic disturbances and water quality with a higher FBI value [19]. This water body was classified as V and IV Classes by the BMWP and SIGNAL, respectively; a similar tendency in India’s wetlands was obtained, with a BMWP value more restrictive than the SIGNAL value [21], and on the contrary, a more restrictive SIGNAL value in another study [22]. This body had the second lowest water quality for aquatic life [5,6].

#### 4.1.3. Chada

Differences in abundance were found both between SSs and seasons. During spring sampling, the only significantly different SS with a greater abundance level was SS4, coincidentally one of the farthest from the main access point and, together with SS3, the one with the least anthropic intervention. However, during fall sampling, no insects were found in SS4. One possible explanation is that this body of water corresponds to an irrigation reservoir, where the volume of water stored varies every certain number of months [53], and a controlled decrease in the water volume in the fall may have altered the establishment of aquatic insects in stations such as SS4. The most frequent family was Corixidae (score 3), which contributes to a “regular” quality class, followed by Chironomidae (score 7). Despite reflecting a “bad” quality class, this family was found in low quantities, not negatively affecting the quality classes. On the other hand, Corixidae, represented by *Sigara* sp., here had a lower EC (<180 µS/cm) value than the previous bodies of water, accounting for the diversity of habitat [42]; it may be a different species of the same genus. The family that contributed most to a “regular” quality class was Baetidae (score 4) [10].

As per the physicochemical variables, it is noteworthy that, in the fall seasons for each SS, the DO concentration decreased, and in no case was there a value equal to or greater than 7, which allows the development of aquatic life [5]. Despite this DO deficiency, aquatic insect families from the Corixidae, Chironomidae, and Baetidae families were also found. It is impossible to determine the same pattern of results in the case of pH since its values ranged between 7.3 and 9.2, with the highest value being found in SS4 in the fall (2018). Regarding EC, although the values fluctuated between 76 and 335 µs/cm, in all cases, a “low” EC rating was considered for the other two water bodies. A possible explanation for these low values may be geographical location; being close to the Andes Mountains, they may receive a lower influence of ion charge because of discharges from anthropic activities. Also, since the geology of an area determines the quantity and type of ions [54], this low EC rating could be determined mainly by geology and geography. Another possible explanation for this temporal variation in the values of the physicochemical variables may be their artificiality, allowing them to be altered in volume and, consequently, in the concentration of the elements that make up the variables above [27]. A global meta-analysis revealed that EC highly correlates with invertebrate richness in artificial reservoirs and DO levels in natural ponds [40]. Also, the phenology of each insect family adjusts to the conditions of DO, pH, EC, and other variables (T, TSS, P, and N) that do not temporarily affect their life cycle. Comparing our FBI results by SSs, the values generally indicated quite different water levels than the physicochemical indices here, coinciding with those obtained from Poland’s lakes [18]. Here, the BMWP and SIGNAL obtained equal water quality (Class V) ratings with each other and were more severe than the FBI rating (Class I); however, the BMWP obtained a lower water quality than the SIGNAL with wetlands in India [21], contrasting with the SIGNAL rating of another study [22]. Regarding the other water bodies, Chada had the best quality for aquatic life [5,6], mainly due to its DO and pH values, the latter with an acceptable level in all SSs, but not due to temperature.

### 4.2. A Global Analysis of the Three Water Bodies

In lentic bodies, the water surface environment is important for insects, significantly influencing their life histories [25]. Although the FBI described deterioration in the water bodies, this was inconsistent with the trends indicated using the standards and agreed with results from a study of Polish lakes [19]. This is despite a study on lakes in the USA that had been selected for their best discriminatory power and the highest stressor correlations within each group [55], but under different conditions. On the contrary, statistically, the other biological indicators differed from the FBI. They had a more stable situation, equaling or degrading the standards in water quality, especially the BMWP, which is more restrictive and thus contributes to protecting the water bodies studied. This trend is shared by another study on wetlands in India [21]. In contrast, another study in India indicated that the SIGNAL is more severe [22], although it was utilised under different conditions from those studied here.

A CCA of the water bodies evaluated generally coincided with the categorization made from the physicochemical variables, with a high degree of compliance in EC, DO, P, and TSS values; however, it presented non-compliance in the temperature, pH, and N levels, with large fluctuations. This may be mainly due to the point sampling carried out on riverbanks on surface waters, where, due to the low depth, the temperature is like that of the environment, which varies between the spring and fall, considering the Mediterranean climate in the central zone of Chile [2]. These results coincide in part with a study of a wetland in India that also obtained correlations with TSS and EC values [21]. Another study of lakes in Polonia also obtained correlations with EC values [18] and a negative correlation with DO values. Still, they obtained positive correlations with DO values [18]. A global meta-analysis revealed that EC had a high correlation with invertebrate richness in artificial reservoirs [40], coinciding with Chada.

Chada was the best-evaluated water body regarding the CCA, which coincided with the categorization from the physicochemical variables. This water body had high EC, DO, and TSS compliance. Through this analysis, the temperature presented a low tendency in the distribution of data; however, it influenced very significantly the families of aquatic insects, determining their metabolism, primary productivity, respiration, and decomposition of organic matter; in addition, the temperature was closely related to DO, since at higher temperatures, DO levels decrease, which affects negatively water quality [54]. For DO, low levels hinder the presence of aquatic life, an indicator of organic matter contamination [56].

From the discussion of the physicochemical variables in Carén, our results generally showed lower EC values than those obtained by the SMA (2020) [51]. However, a greater concentration of the results was evident in DO levels, a situation like that occurring in Chile.

A positive correlation occurred between EC and TSS levels in the four Batuco SSs, markedly separated from the other water bodies. This could be due to the very nature of this body, with high salinity values related to EC, showing a unique situation that is not comparable to Chada or Carén. Along with these two variables, P had a similar trend, which was not reflected with N. However, the most important element to study the eutrophication processes is P, since, unlike N, it does not interact directly with the atmosphere, better reflecting what occurs in water bodies [57]. Likewise, some studies establish that EC, as an indirect measure of the number of soluble ions, can cause phosphate precipitation [58]. Regarding P and pH, there was no relationship between both variables, so these variables appear independent, as determined by Carrera-Villacrés et al. (2018) [59]. Coinciding with the conclusions by González et al. (2024) [60], these findings can have significant implications for water management and decision-making, providing information for implementing appropriate strategies to preserve water quality and ensure water security in the lentic bodies of central Chile.

## 5. Conclusions

The incorporation of aquatic insect families as bioindicators allows ecosystem dynamic results to be evidenced through measurement sequences that reflect the interaction between physicochemical phenomena and anthropic disturbances of these lentic bodies. The eudominant insect families were Corixidae and Chironomidae in Batuco, Chironomidae and Corixidae in Carén, and Corixidae in Chada. Baetidae was dominant in Carén and Chada. The water bodies were classified in descending order of water quality by Chilean standards: Chada > Carén > Batuco. Based on the nitrogen and phosphorus levels in the water bodies, all of them were eutrophic. TSS (total suspended solids), P (phosphorus), and EC (electrical conductivity) were strongly positively correlated and negatively associated with DO (dissolved oxygen). The TSS level was the most significant influential factor. Although the FBI described deterioration in these water bodies, this was not very consistent with the trends indicated by using the standards. On the contrary, statistically, the other biological indicators differed from the FBI. They had a more stable situation, equaling or degrading the standards in water quality, especially the BMWP, which is more restrictive and thus contributes to protecting the water bodies studied. In the case of Chada, this study contributes to the conservation of this lentic body because it corresponds to the first record of its water quality, constituting a basis for future studies. Additionally, a sequence of records of measurements carried out in this area would account for the dynamics of the ecosystem through the different seasons. Considering the diversity of these water bodies, it is possible to determine not only the water quality but also the conservation state of the whole environment.

## Figures and Tables

**Figure 1 insects-15-00831-f001:**
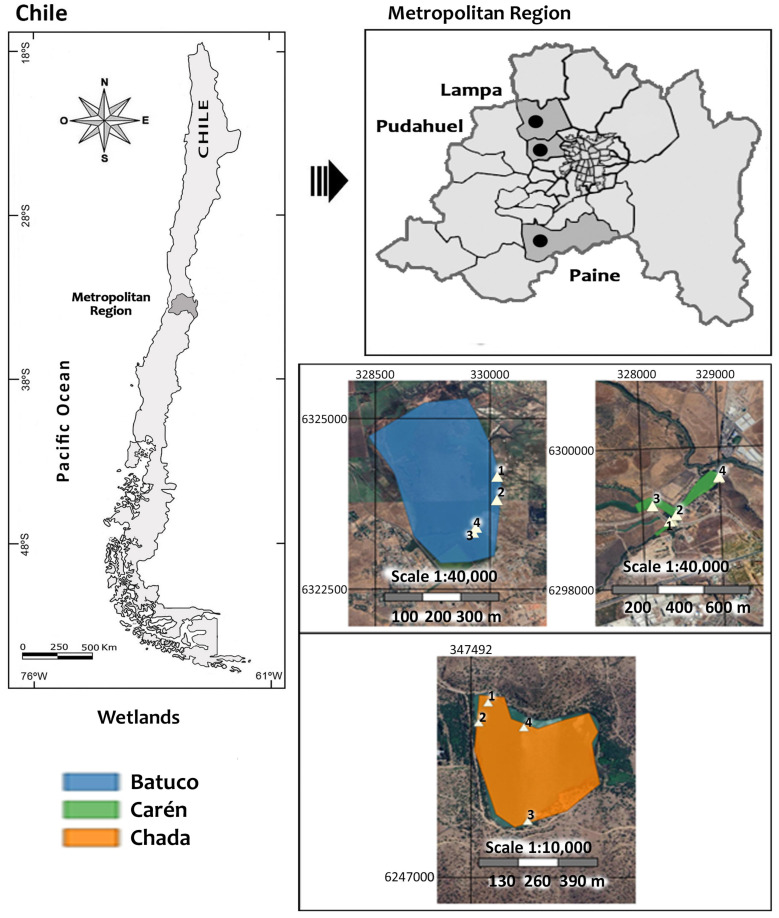
The geographic location of the three lentic bodies studied in the Metropolitan Region (MR), Chile, with their four sampling stations (SS) (modified from Google 2018, Earth Pro ™ version 7.1.5.1557).

**Figure 2 insects-15-00831-f002:**
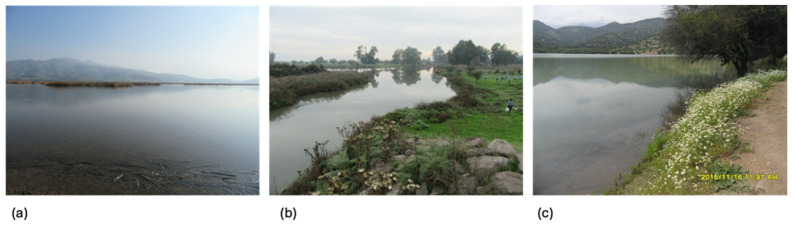
Views of the landscapes of the studies sites in MR of Chile: (**a**) Batuco, (**b**) Carén, and (**c**) Chada.

**Figure 3 insects-15-00831-f003:**
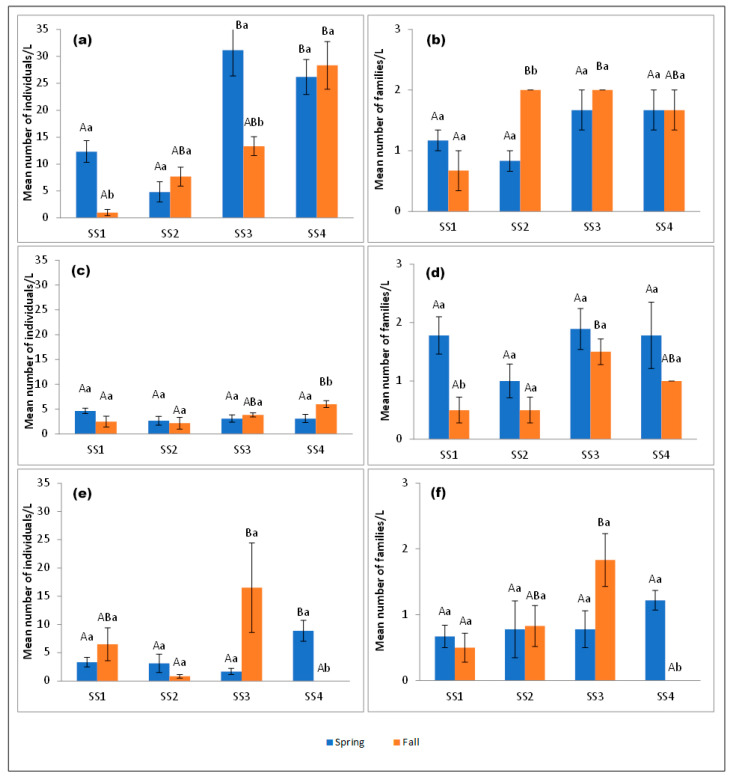
The abundance (mean ± SE) and richness (mean ± SE) levels of aquatic insects in lentic bodies of the MR by season and SS (*n*= 1103 individuals). (**a**) The abundance in Batuco, (**b**) richness in Batuco, (**c**) abundance in Carén, (**d**) richness in Carén, (**e**) abundance in Chada, and (**f**) richness in Chada. SS: sampling station. SE: standard error. Different upper- and lower-case letters indicate significant differences between SSs (Dunn’s tests) and seasons (Mann–Whitney tests) (*p* < 0.05), respectively.

**Figure 4 insects-15-00831-f004:**
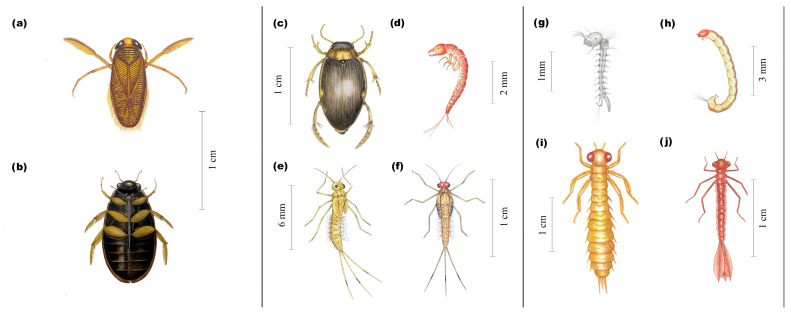
Drawings of insect families identified in the water bodies. (**a**) Corixidae (Hemiptera), dorsal view. (**b**) Hydrophilidae (Coleoptera), ventral view. (**c**) Dytiscidae (Coleoptera) (adult), dorsal view. (**d**) Dytiscidae (larva), lateral view. (**e**) Baetidae (Ephemeroptera) (nymph), dorso-lateral view. (**f**) Baetidae (nymph), dorsal view. (**g**) Culicidae (Diptera) (larva), dorsolateral view. (**h**) Chironomidae (Diptera) (larva), lateral view. (**i**) Aeshnidae (Odonata) (nymph), dorsal view. (**j**) Coenagrionidae (Odonata) (nymph), dorsal view. Courtesy of Carmen Tobar M., scientific illustrator.

**Figure 5 insects-15-00831-f005:**
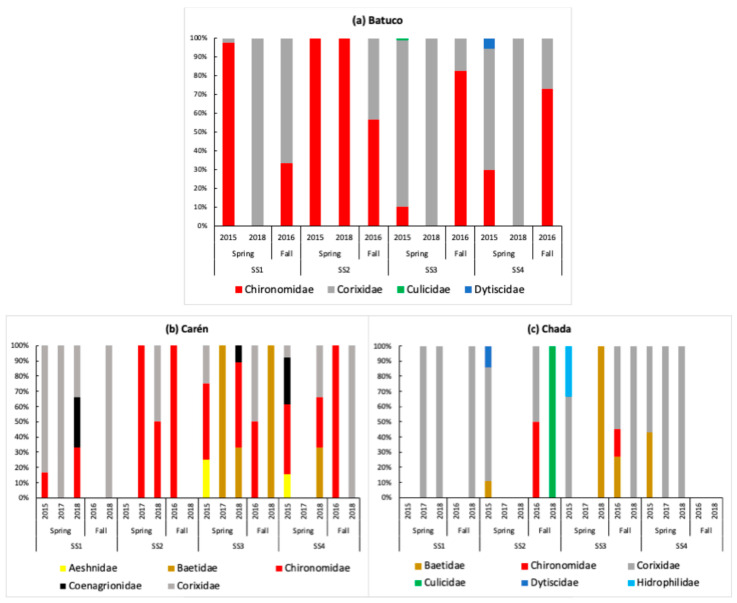
The relative abundance (%) of aquatic insect families by sampling station (SS), season, year and water bodies, MR, Chile. (**a**) Batuco (*n* = 598 individuals), (**b**) Carén (*n* = 209 individuals), and (**c**) Chada (*n* = 296 individuals).

**Figure 6 insects-15-00831-f006:**
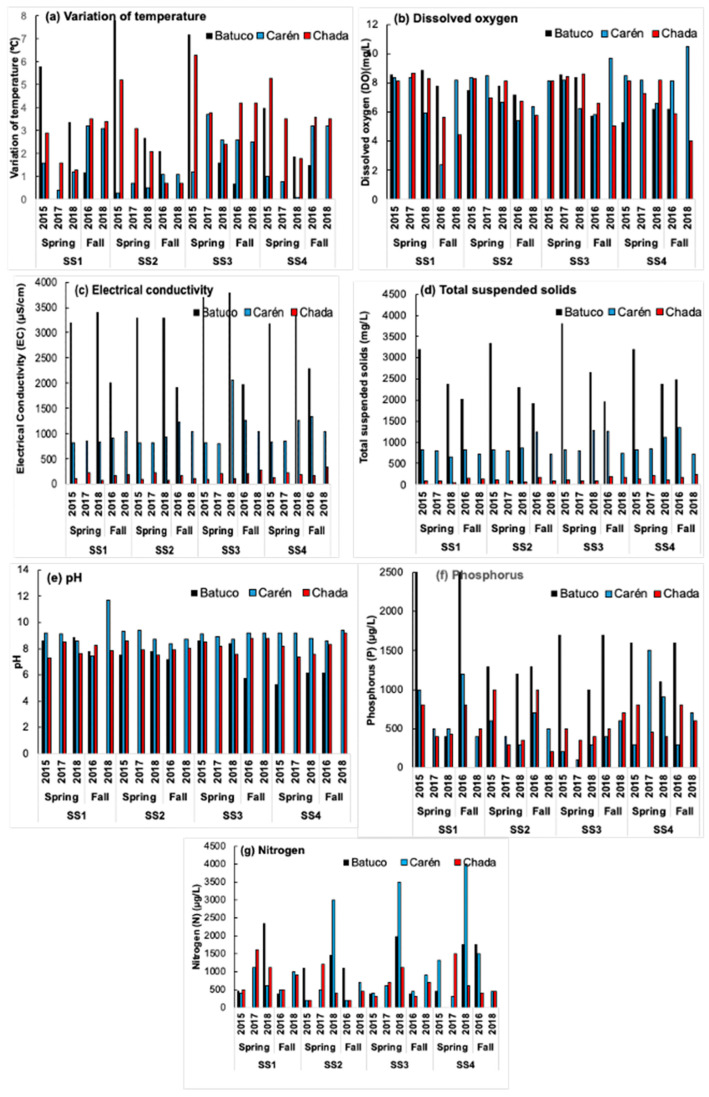
The variation in temperature (T) (**a**), dissolved oxygen (DO) (**b**), electrical conductivity (EC) (**c**), total suspended solids (TSS) (**d**), pH (**e**), phosphorus (P) (**f**), and nitrogen (N) (**g**), in the three water bodies, SS, seasons, and years.

**Figure 7 insects-15-00831-f007:**
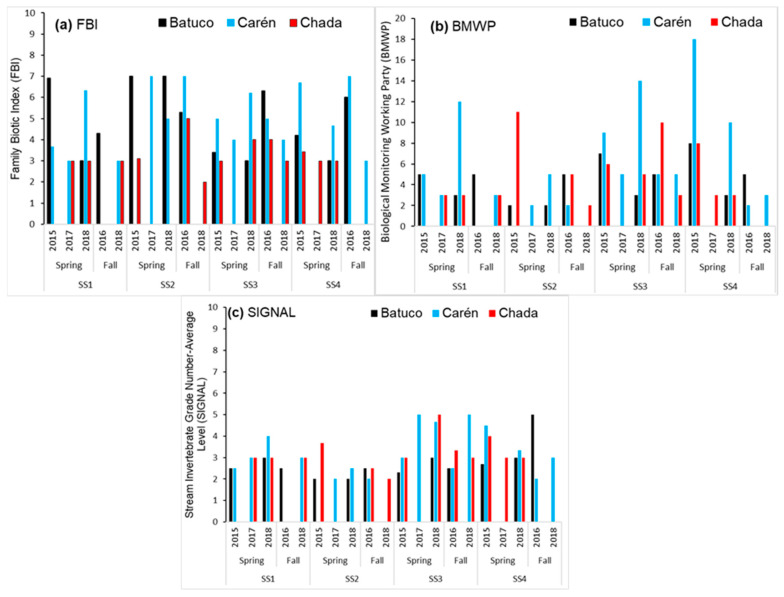
Biotic indices values in the SS, seasons, and years by water bodies, adapted for Chile by Figueroa et al. (2007) [10]. (**a**) The Family Biotic Index (FBI), (**b**) Biological Monitoring Working Party (BMWP), and (**c**) Stream Invertebrates Grade Number-Average Level (SIGNAL).

**Figure 8 insects-15-00831-f008:**
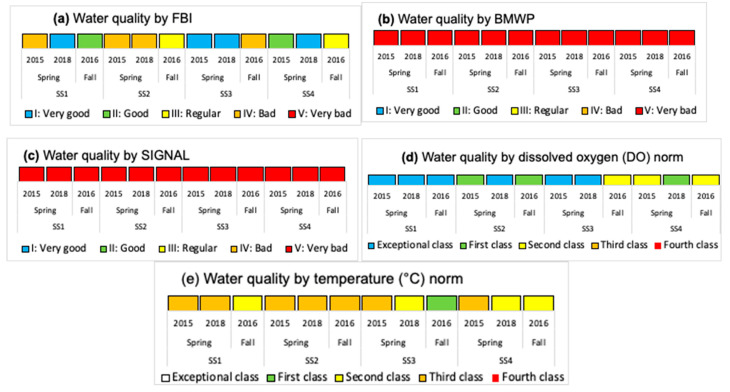
The water quality class classification considering the (**a**) Family Biotic Index (FBI), (**b**) British Biological Monitoring Working Party score system (BMWP), and (**c**) Stream Invertebrate Grade Number-Average Level (SIGNAL), adapted for Chile by Figueroa et al. (2007) [10], and from the Chilean Standard NCh 1.333 [5] and guide [6] with (**d**) dissolved oxygen (DO) and (**e**) temperature (T) in the water bodies, SS, seasons, and years, in Batuco.

**Figure 9 insects-15-00831-f009:**
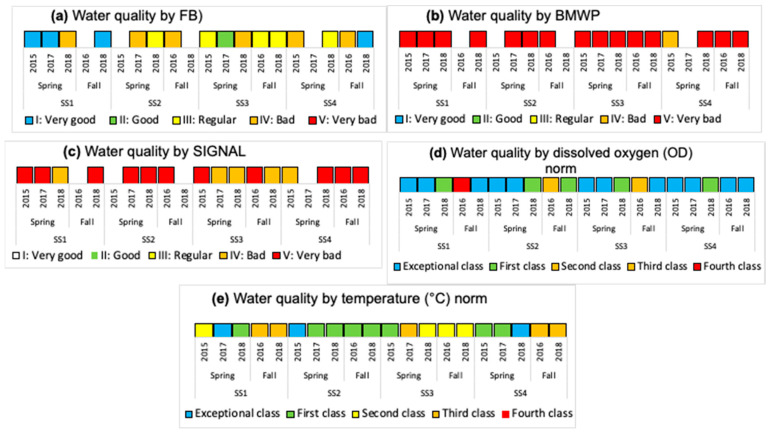
The water quality class classification considering the (**a**) Family Biotic Index (FBI), (**b**) British Biological Monitoring Working Party score system (BMWP), and (**c**) Stream Invertebrates Grade Number-Average Level (SIGNAL) by Figueroa et al. (2007) [10], and from the Chilean Standard NCh 1.333 [5] and guide [6] with (**d**) dissolved oxygen (DO) and (**e**) temperature (T) in the water bodies, SS, seasons, and years, in Carén.

**Figure 10 insects-15-00831-f010:**
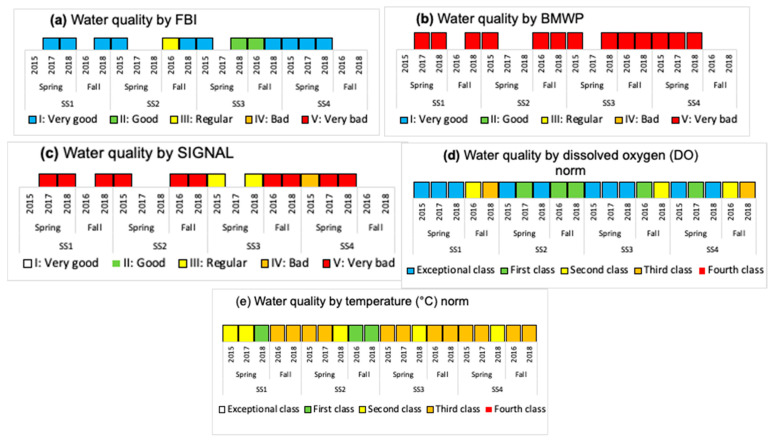
The water quality class classification considering the (**a**) Family Biotic Index (FBI), (**b**) British Biological Monitoring Working Party score system (BMWP), and (**c**) Stream Invertebrates Grade Number-Average Level (SIGNAL) by Figueroa et al. (2007) [10], and from the Chilean Standard NCh 1.333 [5] and guide [6] with (**d**) dissolved oxygen (DO) and (**e**) temperature (T) in the water bodies, SS, seasons, and years, in Chada.

**Figure 11 insects-15-00831-f011:**
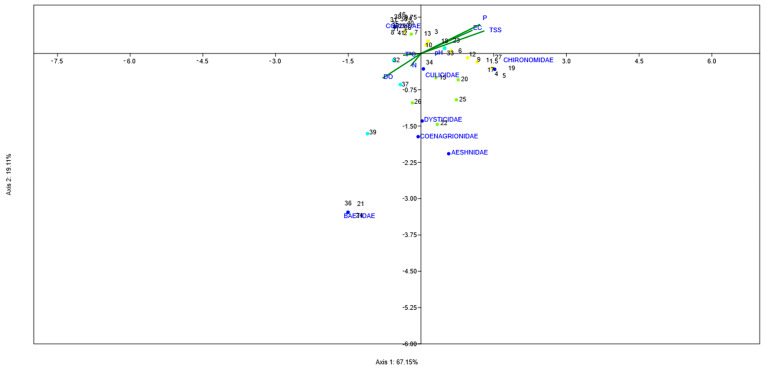
The triplot of the Canonical Correspondence Analysis (CCA) of aquatic insect family assemblages (blue circles) and physicochemical variables (green lines) in the three water bodies during the period of study. Abbreviations used for different observations (numbers) are as follows: Batuco: 1–14 (yellow circles), Carén: 15–28 (green circles), and Chada: 29–41 (calypso circles).

**Table 1 insects-15-00831-t001:** The number of aquatic insect specimens by taxon in water bodies in MR, Chile (2015–2018). The classification of the abundance ranks follows Engelmann (1978) [36].

Taxon	Water Bodies					
	Batuco		Carén		Chada	
	No. of Specimens *	Abundance Rank	No. of Specimens	Abundance Rank	No. of Specimens	Abundance Rank
COLEOPTERA Dytiscidae	4	Recedent	-	-	4	Recedent
Hydrophilidae	-	-	-	-	2	Subrecedent
DIPTERA						
Chironomidae	211	Eudominant	88	Eudominant	18	Subdominant
Culicidae EPHEMEROPTERA	1	Sporadic	-	-	3	Recedent
Baetidae	-	-	26	Dominant	43	Dominant
HEMIPTERA						
Corixidae	382	Eudominant	82	Eudominant	226	Eudominant
ODONATA						
Aeshnidae	-	-			-	-
Coenagrionidae	-	-	10	Subdominant	-	-

* “-“ = not detected.

**Table 2 insects-15-00831-t002:** Water quality classification based on water quality biological indices by water bodies.

Biological Indicators/Class/Significance	Batuco	Carén	Chada
Family Biotic Index (FBI)	4.94	5.10	3.27
Class/Significance	III/Regular	III/Regular	I/Very good
Biological Monitoring Work Party (BMWP)	4.41	6.44	5.00
Class/Significance	V/Very bad	V/Very bad	V/Very bad
Stream Invertebrate Grade Number-Average Level (SIGNAL)	2.54	3.35	3.19
Class/Significance	V/Very bad	IV/Bad	V/Very bad

**Table 3 insects-15-00831-t003:** Values of the total (Friedman) (* *p* < 0.0001) and paired (Wilcoxon) (* *p* < 0.01) comparative tests of the biological indices.

	Friedman Ranking Test			
Biological Indices	*n*	gl	*T* Value	*p* Value
FBI/BMWP/SIGNAL	41	2	390.85	<0.0001 *
	Wilcoxon signed-rank test			
	*n*	*Z* value		*p* value
FBI/BMWP	41	−5.65		0.00 *
FBI/SIGNAL	41	−5.64		0.00 *
BMWP/SIGNAL	41	−2.80		0.01

## Data Availability

The data presented in this study are available on request from the corresponding author.

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
