# Peer review of "Utilizing Insects as Bioindicators: An Approximation for Conservation in Urban Lentic Ecosystems in Central Chile"

_insects, 2024, doi:10.3390/insects15110831_

Round 1
Reviewer 1 Report
Comments and Suggestions for Authors
he manuscript ""Utilizing insects as bioindicators: An approach for conservation in urban lentic ecosystems from Central Chile" is important for the area of knowledge and for water resources management because it evaluates the water quality of 3 reservoirs Three lentic bodies in the MR of Chile: Batuco Wetland, Carén Lagoon and Chada Reservoir, urban lentic ecosystems from Central Chile.
The manuscript provides a lot of important information. Suggestions are attached to the manuscript.

Author Response
Response to Reviewer 1 Comments
|
||
1. Summary |
|
|
Thank you very much for taking the time to review this manuscript. We have taken a look at the comments and are delighted at how constructive and comprehensive the comments are. Your wonderful insight and comments will considerably improve the quality of the manuscript, and for this, we are extremely grateful. We carefully considered all your comments, and the authors unanimously agree with all of your comments. Below are the actions we have taken against each comment/suggestion. |
||
2. Point-by-point response to Comments and Suggestions for Authors |
||
Comments 1: The manuscript provides a lot of important information. Suggestions are attached to the manuscript.
|
||
Response 1: Thank you for pointing this out. We agree with this comment. Therefore, we have greatly modified the manuscript.
|
||
Comments 2: Innovation??? Work using biological variables has been done for a long time. When was the study done? In how many reservoirs/lakes? Did the bioindicators indicate anything? Who were the bioindicators?
|
||
Response 2: Thank you for pointing this out. Agree. We have, accordingly, changed/modified those points. All information has been accordingly included in the abstract. “Evaluations were carried out in parallel in four sampling stations in three water bodies, (Batuco Wetland, Carén Lagoon and Chada Reservoir) in 2–3-year series: spring (2015,2017,2018) and fall (2016, 2018) with three replicates” Lines 26-27. “The ecodominant insect families were: Corixidae and Chironomidae in Batuco, Chironomidae and Corixidae in Carén, and Corixidae in Chada. Baetidae was dominant in Carén and Chada. The water bodies were classified in descending order of water quality: Chada> Carén> Batuco”. Lines 30-33. Comments 3: Dispensable paragraph “Conservation is a mission-oriented discipline where values are central among conservationists and other groups [4]. Studies on the role of values in conservation are located within the broader field of environmental factors. Research in this field has ranged from philosophical to sociological, anthropological, and psychological, exploring the range of environmental aspects from individual to population levels [5].”
Response 3: Thank you for pointing this out. Agree. We have eliminated this paragraph.
Comments 4: There is an extensive list of indexes, ok. And why use the FBI, what is the justification? Response 4: Thank you for pointing this out. Agree. We have included a justification. “In our study, we applied the FBI because it was experienced in lotic conditions, as in the lakes of India [16] and Poland [17], and it was designed for use by technical personnel rather than entomology experts who could be trained in family-level identification. Also aligns with the five water quality classes according to national standards [5], making comparing it easier.” Lines: 113-117. |
||
Comments 5: There is a very extensive characterization of the flooded areas. I suggest this characterization in the study area item and not in the introduction Response 5: Thank you for pointing this out. Agree. We have transferred most of the comments of the water bodies to Study Areas, we have only referred to them to justify them in the objectives. Lines: 163-171; Lines: 178-186; Lines: 189-191.
Comments 6: Comment on some work done focusing on lakes, reservoirs...
Response 6: Thank you for pointing this out. Agree. We have comment on some work done focusing on lakes, reservoirs. Lines: 118-128.
Comments 7: I suggest including aspects of the vegetation, such as figs 2a and 2b Does Chada suffer from anthropic action?
Response 7: Thank you for pointing this out. Agree. We have included aspects of the vegetation (Lines: 194-196) and anthropic action (Lines: 189- 191).
Comments 8: Because? Excepto batuco spring and fall 2018
Response 8: Thank you for pointing this out. Unfortunately, for logistical reasons, we were unable to sample during those periods.
Comments 9: Resultados de abundancia y riqueza Could you start the results by referring to: - which sampling station was the most abundant and the one with the greatest richness - which SS had the least richness - which family was the most abundant and where - finish by indicating whether or not there was a difference between SS and the seasons of the year
Response 9: Thank you for pointing this out. We accepted that suggestion of presentation that allowed us to organize the information, thank you very much. Batuco: Lines: 275-278; Carén (Lines:295-298) and Chada (306-309).
Comments 10: Only two of these orders were identified by genera? Any reason?explanation? Tabla 1.
Response 10: Thank you for pointing this out. Our study focused on aquatic insect families, because of how practical their identification is and therefore it could be used by people who are not experts in entomology and thus know the state of health of a lentic body and was possible applicate the FBI. Lines: 114-115. Unfortunately, when we put only a few that we can to identify, it was confusing. The keys to the recognition of aquatic species that we have in Chile are scarce; we do not know if they are native species, and most of them correspond to immature states. We only comment as an observation on the cases of Sigara sp. and Andesiops peruvianus. We agree that it is necessary to advance to the species level due to the differences detected in the health states of the bodies of water, and to understand their behavior, but this was the first approximation.
Comments 11: Could you comment on which SS has the highest concentration of nutrients, SST... and then describe the results comparing them with NCh 1.333 and Guide 8?
Response 11: We are very grateful for this recommendation because it allowed us to better present and interpret the results. Lines: 331-355.
Comments 12: Discussion por cuerpo de aguas
Regarding insect abundance, SS1 and SS2 had small numbers of individuals/L, coinciding with the areas close to the main access to this water body. In the first samples, an evident anthropic impact was observed in the entire sector close to these SS but not in SS3 and SS4, which were very close to each other and further away from the places with greater anthropic intervention. In these last SS, different taxonomic groups of invertebrates were found, highlighting insects. Regarding family richness, among the significant differences between SS and between seasons, a lower richness stands out in SS1, both in the spring and fall, and in the spring in SS2. Regarding insect abundance, SS1 and SS2 had small numbers of individuals/L, coinciding with the areas close to the main access to this water body. In the first samples, an evident anthropic impact was observed in the entire sector close to these SS but not in SS3 and SS4,
What impact? Response 12: Thank you for pointing this out. Agree. We improved our idea. Lines: 416-429; Batuco (Lines: 434-443; Lines 450-452). Carén (Lines: 461-474; 487-490). Chada (Lines:502-504; Lines: 522-524)
Comments 13: This could be because the insect families found, mainly SS1 and SS2, present a wide range of tolerance to changes in these variables. Regarding the FBI values, it is difficult to establish a pattern that allows associating the SS or seasons with a single quality class. Which families? Genera?
Response 13: Thank you for pointing this out. Agree. We improved our idea. Lines: 422-428.
Comments 14: The Corixidae family, with a score of 5 [14], contributes to a better-quality class, in this case, “regular”. The Corixidae family was represented by Sigara sp., a species of a cosmopolitan genus is further subdivided into many subgenera [30] Sigara sp is only a genus Dispensable
Response 14: Thank you for pointing this out. Agree. We corrected our mistake (Lines: 426) and eliminated the dispensable phrase.
Comments 15: In general, a low average number of individuals/L occurred throughout the seasons in all SS. The average did not exceed 5 inds./L, with no significant differences between SS during the spring and slight differences between SS3 and SS4 in the fall. This low abundance coincides with the characteristics of this water body, with many of its physical and chemical variables not favoring aquatic life development.
What does it mean? What did you mean?
Response 15: Thank you for pointing this out. Agree. We re-write the idea. Lines: 461-465.
Comments 16: Conclusiones give more emphasis to this item. If the objective was to contribute to the knowledge of the dynamics of aquatic insect assemblages as bioindicators of water quality, I think that reference should be made to this quality in the different sampling sites. Was the use of the FBI a good indicator?
Response 16: Thank you for pointing this comment. Agree. We are following your recommendations. We have added those comments. Reference to this quality in the different sampling sites. Lines: 570-574. Was the use of the FBI a good indicator? Lines: 576-579.
|
Reviewer 2 Report
Comments and Suggestions for Authors
The authors have investigated the use of aquatic insects along with physico-chemical sampling to determine the health of three lentic water bodies (ponds/lakes/wetlands) in central Chile. In each water body, sampling was carried out at four stations in winter and summer and in two or three years. Standard chemical sampling was undertaken at the same time as biological sampling. The level of insect identification is coarse (family), which is a problem with Chironomidae in particular, as species have a very wide range of different responses to enrichment, pollution and habitat variables. The manuscript is not easy to follow and could be much more concise. For example, the simple summary is very rambling in style, and the long Introduction is very general apart from the descriptions of the three wetlands which would be better placed in Study areas (Section 2.1) where other aspects of them are considered. Scientifically, I wonder how efficient the biological sampling was. It is stated that samples were taken with a 1 litre container immersed in the water body, but it is not stated where exactly the samples were taken or in what depth of water. Were they taken at the edge of the water bodies in open water, or were they in weed beds? What was the diameter of the sampling container and why would insects be expected to enter it? One litre is very small. Larvae of the odonate and ephemeropteran families found are usually attached to vegetation or living on the bottom sediments and even Sigara and dytiscid beetles grip vegetation much of the time. The question really is why did you not use a net for sampling?
The results are presented in several tables and figures. The drawings of insects in Figure 3 are attractive, but more appropriate for a general or non-specialist audience since the insect families depicted are all well-known to freshwater biologists in general. Figure 5 is a nice set of graphs with appropriate error bars and indicators of statistical significance. I found the legend to Figure 9 hard to understand and wonder whether the PCA is adequately informative to include. The strengths of the water quality variables are presumably given by the lengths of the arrows, which could be pointed out.
The discussion deals with the three water bodies in succession but is essentially an extension of the Results. Furthermore, the results are not discussed in relation to other wetlands, or the literature, as one would expect in an international scientific paper as opposed to a local report.
Comments on the Quality of English LanguageThe written English needs considerable attention, and the various sections of the paper need to be better organised.
Author Response
Response to Reviewer 2 Comments
|
||
1. Summary |
|
|
Thank you very much for taking the time to review this manuscript. We have taken a look at the comments and are delighted at how constructive and comprehensive the comments are. Your wonderful insight and comments will considerably improve the quality of the manuscript, and for this, we are extremely grateful. We carefully considered all your comments, and the authors unanimously agree with all of your comments. Below are the actions we have taken against each comment/suggestion. |
||
2. Point-by-point response to Comments and Suggestions for Authors |
||
Comments 1: The level of insect identification is coarse (family), which is a problem with Chironomidae in particular, as species have a very wide range of different responses to enrichment, pollution and habitat variables. |
||
Response 1: Thank you for pointing this comment. We agree that it is necessary to advance to the species level due to the differences detected in the health states of the bodies of water, and to understand their behavior, but this was the first approximation. Our study focused on aquatic insect families, because of how practical their identification is and therefore it could be used by people who are not experts in entomology and thus know the state of health of a lentic body. Unfortunately, when we put only a few that we can to identify, it was confusing. The keys to the recognition of aquatic species that we have in Chile are scarce, we do not know if they are native species and most of them corresponded to immature states. We only comment as an observation on the cases of Sigara sp. and Andesiops peruvianus. We hope advance in future. We included the following reference as explication “Overall interpretations drawn for large and diverse groups such as Diptera or Chironomidae may be overly simplistic. However, general points are useful to summarise primary associations and the group-level information, with due caution, could be valuable. (Buffagni 2019)”. Lines: 435-438.
|
||
Comments 2: The manuscript is not easy to follow and could be much more concise. For example, the simple summary is very rambling in style, and the long Introduction is very general apart from the descriptions of the three wetlands which would be better placed in Study areas (Section 2.1) where other aspects of them are considered. |
||
Response 2: Thank you for pointing this out. Agree. We have, accordingly, changed/modified those points. All information has been accordingly included in the Abstract (Lines:13-17), in Introduction (Elimination paraph 2, added justification FBI (Lines 113-117; added comments others study in lentic conditions (Lines: 118-128); We have transferred most of the comments of the water bodies description to Study Areas, we have only referred to them to justify them in the objectives. Lines: 163-171; Lines: 178-186; Lines: 189-191. Comments 3: Scientifically, I wonder how efficient the biological sampling was. It is stated that samples were taken with a 1 litre container immersed in the water body, but it is not stated where exactly the samples were taken or in what depth of water. Were they taken at the edge of the water bodies in open water, or were they in weed beds? What was the diameter of the sampling container and why would insects be expected to enter it? One litre is very small. Larvae of the odonate and ephemeropteran families found are usually attached to vegetation or living on the bottom sediments and even Sigara and dytiscid beetles grip vegetation much of the time. The question really is why did you not use a net for sampling? Response 3: Thank you for pointing this out. In relation to why a plastic container and not a net, this is because there are Chilean standards that indicate the use of airtight containers, including polyethylene containers, both for the of water quality study from the physical-chemical point of view as well as for biological studies, in which the study of insects can be included. One liter was chosen so that the volume was comparable both for the parameters and for the study of insects, being said volume used in water analysis in Chile. We included in the manuscript the paragraph “Sampling was done by immersing a 1 L sterile polypropylene jar (8 cm diameter cm, longitude 20 cm) with a hermetically sealed lid, used to quantitatively sample a known area in the water body and remove it once filled, according to Official Chilean Water Quality Standard NCh-ISO-5667/4:2016 [27]. Depending on the complexity, these samples were taken from the edge of the water body with a range between 20-100 cm and superficially from a transversal section of the water body, no more than 50 cm depth”. Lines: 212-217.
Comments 4: The drawings of insects in Figure 3 are attractive, but more appropriate for a general or non-specialist audience since the insect families depicted are all well-known to freshwater biologists in general. Response 4: We include these illustrations because they were made especially for this work by the outstanding Chilean scientific illustrator of several entomology books, Carmen Tobar. We were honored to have your collaboration, and as a sign of gratitude, we included it in the manuscript. |
||
Comments 5: Figure 5 is a nice set of graphs with appropriate error bars and indicators of statistical significance.
Response 5: Thank you for pointing out this comment.
Comments 6: I found the legend to Figure 9 hard to understand and wonder whether the PCA is adequately informative to include. The strengths of the water quality variables are presumably given by the lengths of the arrows, which could be pointed out. Response 6: Thank you for pointing this out. We believe that the PCA is a tool that helps us interpret the data. Thank you very much for the comment about the size of the arrows. We have incorporated it (Lines: 388-390). Comments 7 The discussion deals with the three water bodies in succession but is essentially an extension of the Results. Furthermore, the results are not discussed in relation to other wetlands, or the literature, as one would expect in an international scientific paper as opposed to a local report. Response 7: Thank you for pointing this out. We agree. We have improved the discussion. Batuco: Lines:116-428; Lines: 434-443; Lines: 450-452. Carén: Lines: 467-472; Line: 474; Lines: 487-489; Chada: Lines: 502-502; lines 522-24.
|
Reviewer 3 Report
Comments and Suggestions for Authors
General comments:
This work is important to highlight the importance of aquatic insects as a bio monitoring agent. Authors have taken important lentic systems to study taking different seasons over a period of years. However, I found the study lacks estimation of important water variables and lacks analyzing important statistics like Pearson/Spearman correlation and CCA (see the full form below). Some plots need to restructure or should be made more clear. The discussion need to restructure and add comparison with previous studies on aquatic insects in lentic systems (e.g. ponds, oxbow lake, floodplains etc).
Specific comment:
L16-18 sounds same as L22-24 and L40. make some changes for each lines
L29: in-sects????
L104, remove the word juveniles as some order like Hemiptera and coleoptera has entire life stage aquatic.
Why have you used FBI for bio monitoring calculations? They are meant exclusively for streams i.e., lotic systems, whereas you have done work on lentic systems. Both the systems (lentic and lotic) has different composition of aquatic insects. Therefore, I strongly oppose using FBI, instead I would recommend using BMWP , ASPT where lake systems were considered and SIGNAL2 (which has a mention of lentic systems, though they worked mainly on streams). For BMWP and ASPT follow Mandaville 2002 and for SIGNAL2 follow Chessman 2003.
Where is the dominance status in the study? authors mentioned the word “frequent” in the results to to define most dominant family int the sites. Use Engelmann scale of dominance to see possible dominant aquatic insect family in different systems. This will give the aerial view of the pollution status of the systems based on the dominant family biotic score/index.
I find it very hard to understand fig. 4. what are the different shades signify ? Does the presence of shades indicates highest abundance?
Fig. 6b, correct the spelling of oxygen
Very limited water variables were estimated. Important variables related to aquatic insects are FCO2, total alkalinity, water depth, water transparency using Secchi disk are missing.
Fig. 8 needs more clarity in visualisation.
Instead of PCA I would recommend to use CCA (Canonical correspondence analysis) triplet . This will give better representation of aquatic insect families assemblages in accordance with water variables and sites/seasons.
Why only confined to family level in identification? I know only family level is required for the biomonitoring. But these days, we have well defined keys at least identification till genus level which can help in better prediction in multivariate analyses like PCA or CCA.
Entire discussion need to rewrite. When you did PCA (Better do CCA) you will get the idea what are the important env.variables that is affecting the insect assemblages. You need to focus more on that variables and how/why they are related with insect. Unnecessary explaining each and every variables in details does not make sense. the discussion for each aquatic systems should be to the point highlighting the relationship between aquatic insect family with important variables of water, what the status of the pollution coincided by both.
Also, I didn't find a single comparison of the sites and work with any previous works that worked on similar sites/work.
L366-367: didn't understand the meaning?
Author Response
Response to Reviewer 3 Comments
|
||
1. Summary |
|
|
Thank you very much for taking the time to review this manuscript. We have taken a look at the comments and are delighted at how constructive and comprehensive the comments are. Your wonderful insight and comments will considerably improve the quality of the manuscript, and for this, we are extremely grateful. We carefully considered all your comments. Below are the actions we have taken against each comment/suggestion. |
||
2. Point-by-point response to Comments and Suggestions for Authors |
||
Comments 1: This work is important to highlight the importance of aquatic insects as a bio monitoring agent. Authors have taken important lentic systems to study taking different seasons over a period of years. |
||
Response 1: Thank you for pointing this out.
|
||
Comments 2: However, I found the study lacks estimation of important water variables and lacks analyzing important statistics like Pearson/Spearman correlation and CCA (see the full form below). Some plots need to restructure or should be made more clear. The discussion need to restructure and add comparison with previous studies on aquatic insects in lentic systems (e.g. ponds, oxbow lake, floodplains etc).
|
||
Response 2: Thank you for pointing this out. Below, we will respond to these comments point by point.
Comments 3: L16-18 sounds same as L22-24 and L40. make some changes for each lines Response 3: Thank you for pointing this out. It was a mistake. We have eliminated this paragraph.
Comments 4: L29: in-sects???? Response 4: Thank you for pointing out this observation. Agree. It was a mistake. We have corrected it. |
||
Comments 5: L104, remove the word juveniles as some order like Hemiptera and coleoptera has entire life stage aquatic. Response 5: Thank you for pointing this out. Agree. We have eliminated. Line: 99.
Comments 6: Why have you used FBI for bio monitoring calculations? They are meant exclusively for streams i.e., lotic systems, whereas you have done work on lentic systems. Both the systems (lentic and lotic) has different composition of aquatic insects. Therefore, I strongly oppose using FBI, instead I would recommend using BMWP , ASPT where lake systems were considered and SIGNAL2 (which has a mention of lentic systems, though they worked mainly on streams). For BMWP and ASPT follow Mandaville 2002 and for SIGNAL2 follow Chessman 2003. Response 6: Thank you for pointing this out. We agree that the composition of aquatic insects may be different from lentics and lotics, but we have jobs that have been occupied by the FBI under lentic conditions and we include them now de Poland and India (Lines: 113-114). We also added in the introduction a justification for the use of the FBI (Lines:114-117).
Comments 7: Where is the dominance status in the study? authors mentioned the word “frequent” in the results to to define most dominant family int the sites. Use Engelmann scale of dominance to see possible dominant aquatic insect family in different systems. This will give the aerial view of the pollution status of the systems based on the dominant family biotic score/index. Response 7: Thank you for pointing this out. We accepted that suggestion and used Englemann scale. Lines: 228-233. Tabla 1.
Comments 8: I find it very hard to understand fig. 4. what are the different shades signify ? Does the presence of shades indicates highest abundance?
Response 8: We are very grateful for this comment. We have changed and used the structure of dominance (%) by SS. Table 5.
Comments 9: Fig. 6b, correct the spelling of oxygen Response 9: We are very grateful for this observation. It was a mistake. We have corrected it. Figure 6.
Comments 10: Very limited water variables were estimated. Important variables related to aquatic insects are FCO2, total alkalinity, water depth, water transparency using Secchi disk are missing. Response 10. We are very grateful for this observation. The variables used in this study are defined by Chilean regulations for the study of water quality for the conservation of aquatic communities, both in seawater and in continental waters such as lakes and lagoons. In this sense, the variables in Chile are considered critical for the study of their quality with their respective methodologies. It should be noted that the alkalinity variable was measured through pH; the depth was low (< 50 cm) with high transparency in all the bodies analyzed, so they were not presented in the data.
Comments 11: Fig. 8 needs more clarity in visualisation. Response 11: We are very grateful for this comment. We have improved. Figure 8.
Comments 12: Instead of PCA I would recommend to use CCA (Canonical correspondence analysis) triplet . This will give better representation of aquatic insect families assemblages in accordance with water variables and sites/seasons.
Response 12: We are very grateful for this recommendation. However, we did not use CCA because it was used preliminarily, but correlations were low, and this study was the first approximation. In our case with PCA, greater relationships were detected between the variables analyzed.
Comments 13: Why only confined to family level in identification? I know only family level is required for the biomonitoring. But these days, we have well defined keys at least identification till genus level which can help in better prediction in multivariate analyses like PCA or CCA. Response 13: We are very grateful for this comment. We agree that it is necessary to advance to the species level due to the differences detected in the health status of the bodies of water and to understand their behavior, but this was the first approximation. Our study focused on aquatic insect families, because of how practical their identification is and therefore it could be used by people who are not experts in entomology and thus know the state of health of a lentic body. Unfortunately, when we put only a few that we could identify, it was confusing. The keys to recognizing aquatic species in Chile are scarce; we do not know if they are native species, and most of them correspond to immature states. We only comment as an observation on the cases of Sigara sp. and Andesiops peruvianus. We hope to advance in the future.
Comments 14: Entire discussion need to rewrite. When you did PCA (Better do CCA) you will get the idea what are the important env.variables that is affecting the insect assemblages. You need to focus more on that variables and how/why they are related with insect. Unnecessary explaining each and every variables in details does not make sense. the discussion for each aquatic systems should be to the point highlighting the relationship between aquatic insect family with important variables of water, what the status of the pollution coincided by both. Response 14: We are very grateful for this recommendation because it allowed us to better the discussion. We used PCA because it used CCA preliminarily, but correlations were low, and this study was the first approximate. In our case with PCA, greater relationships were detected between the variables analyzed. Lines: Batuco: Lines:116-428; Lines: 434-443; Carén: Lines: 467-472; Line: 474; Chada: Lines: 502-502.
Comments 15: Also, I didn't find a single comparison of the sites and work with any previous works that worked on similar sites/work. Response 15: We are very grateful for this comment because it allowed us to better the discussion. Batuco: Lines: 450-452. Carén: Lines: 487-489; Chada: lines 522-24.
Comments 16: L366-367: didn't understand the meaning? Response 16: We are very grateful for this comment. That phrase was removed because it was confusing. |
Round 2
Reviewer 2 Report
Comments and Suggestions for Authors
I note substantial changes have been made to the manuscript which is noticeably improved as a result. However, the standard of written expression is variable, and some sections need further revision. Thank you for your comments on the sampling method I queried. I note that it is a standard method recommended in Chile and read the preprint by Gonzalez which indicates that and points out its limitations.
Here are some other points to consider:
Simple summary
Should the first sentence be deleted? I think so.
Line 113. Rather than “experienced” I suggest you say ...has been used in lentic conditions... Note I think you mean lentic not lotic here.
Line 118. There is a huge literature on lentic environments particularly lakes. Many of these deal with plankton and fish rather than insects which are largely benthic or littoral. Nevertheless, there has been a lot of work on lake insects in North America and Europe.
Study areas: The additions made to the descriptions of study areas are very good. Perhaps the writer of theses sections could edit other sections of the manuscript that are not as clearly written.
Line 176. Insert “vertebrate” before fauna
Line 213. You mean length 20 cm, not longitude.
Line 213. I assume the lid is closed after the sample has been taken!!
Lines 229-235. The addition is hard to follow. Recast.
Table 1. I think you mean recedent, the term used by Engelmann 1978. How does it differ from sporadic?
Line 298. Baetide (spelling)
Figure 5 legend. Rather than Structure of domination I suggest Relative abundance (%), the term conventionally used.
Lines 331-355. These paragraphs need to be rewritten.
Comments on the Quality of English LanguageThe language still need improvement in some sections as indicated in my report.
Author Response
Response to Reviewer 2 Comments (Round 2)
|
||
1. Summary |
|
|
Again, thank you for taking the time to review the manuscript and give us another chance. We have looked at the comments and are delighted at how constructive and comprehensive the comments are. Your excellent insight and comments will considerably improve the manuscript's quality, and we are extremely grateful. We carefully considered all your comments, and the authors unanimously agreed with all your comments. Below are the actions we have taken against each comment/suggestion. |
||
2. Point-by-point response to Comments and Suggestions for Authors |
||
Comments 1: I note substantial changes have been made to the manuscript which is noticeably improved as a result. However, the standard of written expression is variable, and some sections need further revision. |
||
Response 1: Thank you for pointing out this comment. We appreciate this comment because it made us work to write better. There are substantive changes because we accepted some suggestions by reviewer 3, who recommended that we try other biological indicators and compare them. We include new tables: a) Table 1, with the FBI plus two other indices (BMWP and SIGNAL), already adapted to the Chilean, and b) Table 2, non-parametric tests by Friedman and Wilcoxon to detect differences between the indices. The old Figure 8 was transformed, adding the other two biological indices but only for Batuco. Figure 9 was made for Carén, and Figure 10, Chada. This is to facilitate comparison. Finally, a new figure was added (Figure 11), replacing the previous one with the PCA, where a canonical correspondence analysis (CCA) with physicochemical variables and insect families was now incorporated. We incorporate new comments/rewrites in Simple Summary (Lines: 14-17), Abstracts (Lines: 14-17; 31-35; 38-42), Introduction (Lines:105-106;109-116;119-120); methods (Lines: 212-214; 237-245;248-251;275-280), results (Lines: 339-343;353-387;427-435;439-452), discussion (Lines: 484-486; 501-507;518-520;543-551;581-583;585-590;596-606;613-620;642-646) and conclusions (Lines: 656-664).
We think this improves data comprehension. New references were also incorporated. |
||
Comments 2: Thank you for your comments on the sampling method I queried. I note that it is a standard method recommended in Chile and read the preprint by Gonzalez which indicates that and points out its limitations. |
||
Response 2: We appreciate your words. And that it has allowed us to find González's work, already published and we were able to cite it. Lines: 641, 818-819. |
||
Comments 3: Simple summary. Should the first sentence be deleted? I think so.
|
||
Response 3: Thank you for pointing this out comments. Agree. We have eliminated.
|
||
Comments 4: Line 113. Rather than “experienced” I suggest you say ...has been used in lentic conditions... Note I think you mean lentic not lotic here. |
||
Response 4: Thank you for pointing these observations. It was a mistake, our apologies. We have changed by “in lentic conditions” and eliminate “experienced”. |
||
Comments 5: Line 118. There is a huge literature on lentic environments particularly lakes. Many of these deal with plankton and fish rather than insects which are largely benthic or littoral. Nevertheless, there has been a lot of work on lake insects in North America and Europe.
|
||
Response 5: Thank you for pointing out this comment. Agree. We were able to cite other works on lake insects in North America and Europe. Lines: 109-116; 118-119.
|
||
Comments 6: Study areas: The additions made to the descriptions of study areas are very good. Perhaps the writer of theses sections could edit other sections of the manuscript that are not as clearly written. |
||
Response 6: Thank you for pointing this out. We try to improve and rewrite several paragraphs.
|
||
Comments 7: Line 176. Insert “vertebrate” before fauna.
|
||
Response 7: Thank you for pointing this out. We agree. We have included “vertebrate”. Lines: 175.
|
||
Comments 8: Line 213. You mean length 20 cm, not longitude.
|
||
Response 8: Thank you for pointing this observation. It was a mistake, our apologies. We have changed by “length”. Lines: 212.
|
||
Comments 9: Line 213. I assume the lid is closed after the sample has been taken!! |
||
Response 9: Thank you for pointing this observation. It was a mistake, our apologies. We have rewritten the paragraph. Lines: 212-214. |
||
Comments 10: Lines 229-235. The addition is hard to follow. Recast. |
||
Response 10: Thank you for pointing out this observation. Agee. We have rewritten the paragraph. Lines: 228-233. |
||
Comments 11: Table 1. I think you mean recedent, the term used by Engelmann 1978. How does it differ from sporadic? |
||
Response 11: Thank you for pointing this observation. It was a mistake, our apologies. We have changed by “recedent” in all text and Table 1. Lines: 232-233.
|
||
Comments 12: Line 298. Baetide (spelling)
|
||
Response 12: Thank you for pointing this observation. It was a mistake, our apologies. We have changed by “Baetidae”. Line: 331.
|
||
Comments 13: Figure 5 legend. Rather than Structure of domination I suggest Relative abundance (%), the term conventionally used.
|
||
Response 13: Thank you for pointing out this observation. Agree. We have changed it to “Relative abundance (%)”. |
||
Comments 14: Lines 331-355. These paragraphs need to be rewritten.
|
||
Response 14: Thank you for pointing out this observation. Agee. We have rewritten the paragraph. Lines: 353-397.
|
||
Comments 15: The language still need improvement in some sections as indicated in my report |
||
Response 15: Thank you for pointing out this observation. Agee. We have rewritten several paragraphs. We incorporate new comments/rewrites in Simple Summary (Lines: 14-17), Abstracts (Lines: 14-17; 31-35; 38-42), Introduction (Lines:105-106;109-116;119-120); methods (Lines: 212-214; 237-245;248-251;275-280), results (Lines: 339-343;353-387;427-435;439-452), discussion (Lines: 484-486; 501-507;518-520;543-551;581-583;585-590;596-606;613-620;642-646) and conclusions (Lines: 656-664).
|
||
|
Reviewer 3 Report
Comments and Suggestions for Authors
Authors have addressed most of my comments.
Some additional comments:
I don’t understand the meaning of the word “ecodominant”? I googled it and didn’t find any such word. Replace with word like eudominant or simply dominant.
L114: replace lotic with lentic.
I still insist on adding indices BMWP and ASPT with FBI. Your justification of only using FBI citing reference from Poland (Obolewski et al., 2014) used BMWP and ASPT apart from FBI and stated that BMWP was best suited index among all the indices used. On the contrary, I can show you much latest published works on lentic systems that used BMWP, ASPT, SIGNAL2 and not FBI (Dalal and Gupta 2016, 2018; Choudhury and Gupta, 2017).
Author Response
Response to Reviewer 3 Comments (Round 2)
|
||
1. Summary |
|
|
Again, thank you for taking the time to review the manuscript, insist on your position, and give us another chance. We have analyzed the comments and are delighted with how constructive they are. His wonderful insight and comments greatly improved the quality of the manuscript, and we are incredibly grateful. We carefully considered all of your comments, and the authors unanimously agreed. Below are the actions we have taken regarding each comment/suggestion. |
||
2. Point-by-point response to Comments and Suggestions for Authors |
||
Comments 1: Authors have addressed most of my comments. |
||
Response 1: Thank you for pointing out this comment. We appreciate this comment because it made us work to write better and interpret the results. |
||
Comments 2: I don’t understand the meaning of the word “ecodominant”? I googled it and didn’t find any such word. Replace with word like eudominant or simply dominant.
|
||
Response 2: We appreciate your comment. It was a mistake; our apologies. We have changed by “eudominant” and eliminate “ecodominant” in all text. Lines: 14-35; 471, 521,522, 650. Comments 3: Simple summary. Should the first sentence be deleted? I think so. Response 3: Thank you for pointing this out comment. Agree. We have eliminated. Comments 4: L114: replace lotic with lentic. Response 4: Thank you for pointing this observation. It was a mistake; our apologies. We have changed to “lentic” and eliminated “lotic”. |
||
Comments 5: I still insist on adding indices BMWP and ASPT with FBI. Your justification of only using FBI citing reference from Poland (Obolewski et al., 2014) used BMWP and ASPT apart from FBI and stated that BMWP was best suited index among all the indices used. On the contrary, I can show you much latest published works on lentic systems that used BMWP, ASPT, SIGNAL2 and not FBI (Dalal and Gupta 2016, 2018; Choudhury and Gupta, 2017). Response 5: Thank you for pointing out these comments. There are substantive changes because we accepted all suggestions, and we tried other biological indicators and compared them and discussed. We include new tables: a) Table 1, with the FBI plus two other indices (BMWP and SIGNAL), already adapted to the Chilean, and b) Table 2, non-parametric tests by Friedman and Wilcoxon to detect differences between the indices. The old Figure 8 was transformed, adding the other two biological indices but only for Batuco. Figure 9 was made for Carén and Figure 10, Chada. This is to facilitate comparison. Finally, a new figure was added (Figure 11), replacing the previous one with the PCA, where a canonical correspondence analysis (CCA) with physicochemical variables and insect families was now incorporated. We think this improves data comprehension. New references were also incorporated (Dalal and Gupta 2016, 2018; Choudhury and Gupta, 2017). These analyses transformed our results and showed that the other indices were better, especially BMWP, since it was the most restrictive. We incorporate new comments/rewrites in Simple Summary (Lines: 14-17), Abstracts (Lines: 14-17; 31-35; 38-42), Introduction (Lines:105-106;109-116;119-120); methods (Lines: 212-214; 237-245;248-251;275-280), results (Lines: 339-343;353-387;427-435;439-452), discussion (Lines: 484-486; 501-507;518-520;543-551;581-583;585-590;596-606;613-620;642-646) and conclusions (Lines: 656-664).
|
Round 3
Reviewer 3 Report
Comments and Suggestions for Authors
L110: replace SIGN with SIGNAL. Overall, the present version should be accepted for publication.